# OpenReview forum: "SAFETY-GUIDED FLOW (SGF): A UNIFIED FRAMEWORK FOR NEGATIVE GUIDANCE IN SAFE GENERATION"
_ICLR.cc/2026/Conference — ICLR 2026 Oral_

### Official Review · Reviewer_h63Q · 2025-10-25

**Soundness:** 3
**Presentation:** 3
**Contribution:** 4
**Rating:** 10
**Confidence:** 5

**Summary:**

This paper proposes Safety-Guided Flow (SGF), a novel framework that uses Maximum Mean Discrepancy (MMD)-based potentials within a negative guidance framework for diffusion and flow models to produce safe samples. The authors make two primary theoretical contributions. First, they formally prove that the gradient of the kernel MMD potential is equivalent to the repulsive fields used in Shielded Diffusion and Safe Denoiser, thereby suggesting a unified probabilistic framework. Second, the paper provides evidence for the effectiveness of imposing negative guidance in the initial denoising stage by adopting the control barrier theorem. The effectiveness of the proposed method is validated through comprehensive experiments against both training-based and training-free baselines. The evaluation assesses performance on safety and diversity metrics, along with the ablation studies that confirm the theoretical findings regarding the time window of the negative guidance. The experiments demonstrate substantial safety improvements while having a minimal impact on image quality on benign datasets.
This paper shows its algorithmic significance by proving the existence of a critical time window of negative guidance by proposing a unified probabilistic framework for image generation applying Shielded Diffusion and Safe Denoiser, topics which were previously only dealt with empirically without a theoretical approach. Furthermore, the paper is well structured with systematic and clearly described proofs, supported by extensive experiments.

**Strengths:**

The contributions of the paper are (1) its outstanding theoretical contribution and (2) its novel application of control theory to provide the first formal safety guarantees in this domain.
Strong theoretical foundation: The paper rigorously connects previously empirical safety methods (Shielded Diffusion, Safe Denoiser) through an elegant MMD-potential framework.
Novel control-theoretic insight: The use of the control barrier theorem to justify time-varying safety guidance represents an innovative cross-disciplinary contribution, providing the first formal safety guarantees for generative flows.
Clarity and structure: Proofs and derivations are logically organized, with intuitive interpretations. The writing effectively bridges probabilistic modeling and control theory.
Comprehensive experiments: Evaluations include both safety (nudity suppression, memorization) and generative diversity metrics, supporting the theoretical claims.
Empirical alignment with theory: Ablation results clearly demonstrate the effectiveness of early-stage negative guidance, validating the proposed “critical window” theory.

**Weaknesses:**

The theoretical analysis assumes a weak base drift near the boundary layer, which is reasonable for late denoising stages but could be complemented by a short discussion on its general applicability across different model configurations. The computation of the MMD potential may introduce some inference overhead, though it is likely negligible compared to the overall generation time. While current experiments focus on visual safety domains such as nudity suppression and memorization, extending the framework to more abstract or semantic safety concepts (e.g., copyright or bias) would be a valuable direction for future work, and a brief discussion on this point would further strengthen the paper.

**Questions:**

Regarding the theory, clarification is needed on the following points:
- In Section 4.4, the proof relies on the strong assumption that the base drift \tilde{f} in the boundary layer has a small effect, which is reasonable in the generative model. While the paper reasonably argues that this drift is weak in the final stages, it does not sufficiently address the strength of the gradient of E. The magnitude of the MMD-based guidance depends on the density of the unsafe set. I wonder whether a certain prompt (especially planned adversarial) or the specific data distribution of the unsafe set could lead to cases where \tilde{f} overpowers the safety guidance, i.e., the gradient of E. An adversarial prompt can intentionally steer \tilde{f} (even if small in the final stage) with high precision toward a sparse region in the data distribution of the unsafe set where the magnitude of the gradient of E would also be minimal. Could there be such cases?
- In Section 4.4, a decreasing weight is defined, and the authors interpret this to mean that earlier safety guidance is better. While this is intuitive for standard diffusion, it would be beneficial to specify the exact conditions for this assumption to hold. Are there any non-standard flow models or schedules where this assumption might be violated?
Regarding the experiments, the following should be addressed:
- It seems SGF may introduce an inference cost at sampling time due to the calculation of the MMD gradient at each step. It would be beneficial for the readers to see how this cost scales with the size of the unsafe set, and a comparison with other baselines in terms of inference cost.
- In addition to the comment regarding sparse regions in the data distribution of the unsafe set that may lead to a minimal effect from ||gradient of E||, it seems that the performance of SGF may depend on the quality and size of the unsafe set. It would have been better to also show a sensitivity analysis regarding the quality and size of the unsafe set.
- Does SGF also show strong performance in domains other than nudity? For example, some critical safety concerns in image generation are conceptually abstract and not visually clustered like nudity, such as copyright infringement, or complex harmful concepts. While the MMD potential measures the distance in a feature space and shows effectiveness in capturing negative guidance when dealing with nudity prompts, it may fail to deal with abstract concepts where the unsafe set is difficult to define. Does the MMD potential-based guidance prove effective in preventing this form of conceptual replication?
Minor comments:
1. Page 9: The second paragraph in 5.4 ABLATION STUDIES, there’s a typo showing ‘later times later times’

---

> ### Author Response · Authors · 2025-11-21
> **Official Comment by Authors**
>
> We sincerely appreciate the reviewer’s positive evaluation of our work. We have carefully considered all comments and would like to address the concerns raised.
>
> > ### **Q1 — Can $\tilde f$ overpower the safety guidance $\nabla E$ when the unsafe set is sparse. For instance, adversarial prompts steering to regions with tiny $\|\nabla E\|$?**
>
> This is precisely the edge case excluded by Assumption 1(a), which requires a nondegenerate and well aligned guidance field in the boundary layer. Our choice of the MMD distance helps here, especially when the unsafe set is sparse. It is because $\|\nabla E\|$ is most affected by the nearest unsafe sample, say $y_j$, and far away unsafe samples have small effects, due to exponential decay of the kernel funciton $k_\sigma$; that is, $\|\nabla E\| \approx \frac{2}{N} \| \nabla k_\sigma(\cdot, y_j)\|$. Therefore steering to near an unsafe data pont will be encourtered by the safty guidance $\nabla E$ that maintains a cetain nontrivial magnitude.
>
> Empirically, the MMA Diffusion benchmark is designed to stress exactly this situation and is the most difficult out of our 3 nudity benchmarks, since its prompts are intentionally constructed to elicit very strong nudity and are therefore hard to defend. Even with SGF, the attack success rate (ASR) remains high for some pipelines. For instance, SAFREE + SGF attains ASR = 0.423 on MMA Diffusion, while still providing large relative reductions compared with the corresponding baselines, such as reducing the ASR of SLD from 0.881 to 0.297.
>
> On the other hand, when we examine benign cases, our experiments show that the proposed method behaves as desired. Across benign datasets, SGF consistently achieves substantial safety improvements while having only a minimal impact on image quality, which matches the intended design of the method.
>
> > ### **Q2. — “Earlier is better”: what exact conditions make the decreasing‑weight conclusion hold, and can certain flows/schedules violate it?**
>
> The decreasing weight conclusion is a mathematical consequence of our forward-time dynamics model (Equation (8) in the paper). From the dynamics it follows that the guidance schedule $\beta$ is more influential at an earlier time. Also, requiring less $\beta$ at a later time relies on our Assumption 1, especially (b); this assumption is natural for the situations where the drift $\tilde f$ diminishes near the end of the flow. As the reviewer mentioned, this holds in image based diffusion models that are commonly used for frontier image generation; the magnitude of the denoising updates typically becomes smaller as the process approaches the data manifold. Our theoretical result about earlier guidance being more effective is derived under exactly this type of schedule.
>
> However, there are diffusion language models where these conditions do not hold. Recent works [2-4] on masked diffusion LLMs aims to reduce inference cost while preserving final performance, by changing the unmasking pattern over time. In many of these acceleration methods, the model starts with very conservative unmasking in the early steps and then increases the number of unmasked tokens later, so the effective update size can grow in the later part of the trajectory. This is the opposite trend from the standard image and video schedules that we consider. In such cases, the assumptions used in our theorem are violated, and one would need a more general analysis tailored to these acceleration schedules in order to obtain a rigorous justification.
>
> In contrast, for image and video generation, both our experiments and the reviewer’s understanding rely on the usual schedulers whose step sizes and effective drift magnitudes decrease over time. In this regime, the theoretical analysis in our paper is well aligned with the practical sampling behavior, and the conclusion that earlier safety guidance is preferable is consistent with both the assumptions and the empirical ablations.

---

> ### Author Response · Authors · 2025-11-21
> **Official Comment by Authors (2)**
>
> > ### **Q3 — How does SGF’s inference cost scale with the size/quality of the unsafe set, and how does it compare to other baselines?**
>
> | Models                          | Time (s/img) |
> |---------------------------------|--------------|
> | SD-v1.4                         | 3.18         |
> | + SafeDenoiser (N = 515)        | 3.20         |
> | + Ours (N = 515)                | 3.22         |
> | SAFREE                          | 4.22         |
> | + SafeDenoiser (N = 515)        | 4.24         |
> | + Ours (N = 515)                | 4.32         |
> | + SafeDenoiser (N = 3,200)      | 4.29         |
> | + Ours (N = 3,200)              | 4.70         |
>
> On SD v1.4, SGF with N=515 increases time from 3.18 to 3.22 seconds per image, while Safe Denoiser with N=515 increases time to 3.20. On SAFREE, SGF adds 0.10 seconds for N=515 and 0.48 seconds for $N=3,200$, whereas Safe Denoiser adds 0.02 and 0.07 seconds respectively. We find that the absolute overhead remains as the 1.04 second increase raised by from SD v1.4 to SAFREE. The detailed explanation is updated in the revised manuscript.
>
> For SGF, the additional inference cost is modest and scales in a simple way with the size of the unsafe set. We do not backpropagate through the MMD potential using automatic differentiation. Instead, we use the analytic gradient of the squared kernel MMD in Equation (7), which is a weighted sum of radial kernel terms and is therefore straightforward to implement as a batched reduction on the GPU. As a result, each negative guidance call has complexity that is linear in $|D^-|$, depends only on the number of unsafe exemplars, and is naturally parallelizable.
>
> In practice, as our PyTorch implementation on the kernel and distance computations is supported by GPU parallelization, and the dominant cost in sampling remains the UNet or flow forward passes. This means that the overall runtime overhead introduced by SGF is small.
>
> > ### **Q4 — Sensitivity to the size and quality of $D^-$.**
>
> For this question, our starting point is that the sensitivity to the size and quality of $D^-$ has already been carefully studied in the ablation experiments of Safe Denoisers, in particular in Figure 5(a). Since our method recovers the Safe Denoiser, we expect the same qualitative trend to hold here as well. In that study, when the number of negative samples is reduced, the attack success rate increases, which indicates that it is important for the negative set to be large and diverse enough to cover the unsafe distribution in a meaningful way.
>
> This dependence on the data is not unique to SGF. Most defence methods that rely on data driven signals, including learned pre-filter and post-filter approaches, require sufficient and representative datapoints in order to learn or apply effective safety functions. In this sense, the need for a reasonably rich negative set is a general limitation shared by defence methods and safe generation systems, rather than a specific drawback of our framework. We will add to the appendix a sensitivity plot that varies $|D^-|$ and a simple quality control procedure, for example removing low density outliers, to make this dependence explicit.
>
> From a theoretical viewpoint, our Assumption 1 in Section 4.4  is not realistic when the negative sample size is small, not effectively representing the unsafe set. We believe a more careful theoretical sensitivity analysis will be a good future direction.

---

> ### Author Response · Authors · 2025-11-21
> **Official Comment by Author (3)**
>
> > ### **Q5 — Does SGF work beyond nudity for abstract harms such as copyright or complex harmful concepts?**
>
> Our framework is fully compatible with abstract and complex harmful concepts, and this is already demonstrated in the Safe Denoisers [1]. This work includes experiments on multiple and compositional harmful categories and reports inappropriate generation probability in its Table 3. Since our SGF formulation exactly recovers the Safe Denoiser potential under the corresponding setting, we expect the same qualitative behavior and performance when the same unsafe exemplars and evaluation protocol are used.
>
> Moreover, Safe Denoisers provides concrete examples illustrating intellectual control. They present a case where the text prompt doesn’t mention *Munch* at all, yet the generated image still retains the style of *The Scream*. To eliminate this style, they employ four distinct versions of *The Scream* as negative exemplars. We replicate this experiment and present the results in Appendix F.2 of the revised manuscript. In short, we find that our method generates sharp Barbie portraits with preserved backgrounds while avoiding Munch’s distinctive style.
>
> > **Minor.**
>
> We correct the typo in Section 5.4 “later times later times” to “later times.”.
>
> **Reference**
>
> [1] Kim, Mingyu, et al. "Training-free safe denoisers for safe use of diffusion models." NeurIPS 2025.
>
> [2] Ben-Hamu, Heli, et al. “Accelerated Sampling from Masked Diffusion Models via Entropy Bounded Unmasking.” NeurIPS 2025.
>
> [3] Kim, Seo Hyun, et al. “KLASS: KL-Guided Fast Inference in Masked Diffusion Models.” NeurIPS 2025.
>
> [4] Luxembourg, Omer, Haim Permuter, and Eliya Nachmani. “Plan for Speed–Dilated Scheduling for Masked Diffusion Language Models.” arXiv 2025.

---

> > ### Comment · Reviewer_h63Q · 2025-11-28
> >
> > All my concerns are addressed well. It would be good to include them in the revised paper and the supplement. I already gave the highest score.

---

> ### Author Response · Authors · 2025-12-02
>
> We sincerely appreciate your recognition of our paper and of the efforts we made in revising the manuscript and addressing the raised concerns. It has been a great pleasure to engage with you during the discussion period in deepening our understanding of this study’s significance.
>
> We have carefully incorporated the points raised in the discussion into the revised manuscript. Specifically, Q1 is now addressed in subsection D.4. We created a new subsection in Appendix to cover Q2 and Q4. The material related to Q3 has been added to Table 4 in the main text, and our response to Q5 is now included in subsection G.2 of Appendix.
>
> Once again, we are grateful for your positive assessment of the paper and for your view that this work is suitable for highlights as ***oral or spotlight*** at this venue.

---

### Official Review · Reviewer_UsAs · 2025-10-30

**Soundness:** 2
**Presentation:** 2
**Contribution:** 2
**Rating:** 4
**Confidence:** 4

**Summary:**

The authors propose a unified framework for safety guidance in generating images by using the gradients of an MMD metric to steer the sampling trajectory away from unsafe images. They then perform experiments showing that this strategy works and perform theoretical analysis to show that this work subsumes some prior works.

**Strengths:**

* Tackles an important problem
* Section 4.2 and 4.3 shows that the proposed method subsumes prior work.
* Section 4.4 presents an interesting analysis.
* Empirical studies seem appropriate in illustrating the effectiveness of the proposed approach.

**Weaknesses:**

## Primary concerns
1. How do you pick $s_c$? One of the criticisms of Safe Denoiser is that they pick the interval to apply guidance on heuristically. Is this not the same?
2. What is the compute cost of estimating the MMD and likewise autograd cost for calculating the gradient wrt $\boldsymbol x$? This seems like it could become very expensive once as the size of the unsafe reference dataset grows. How large does $\mathcal D^-$ need to be for the distance to work well, clearly a degenerate singleton distribution should work poorly, right?
3. If we have the $h$ control-barrier function in Section 4.4 why bother with MMD? Wouldn't it be easier to use standard gradient guidance *a la* [1-7] with $h$ instead of calculating the MMD.
4. The **largest** weakness in this paper in my mind is that wouldn't standard gradient guidance with the *control-barrier* function work just as well? I feel like any one of the strategies from [1-7] (there are more papers on this topic but I just listed a few notable ones. For a more complete list of such methods I refer the authors to [2, Figure 5]). If the authors can successfully argue why MMD (or really any probability distance) is more useful for the end goal of safe generation than just using standard gradient guidance with the *control-barrier* function I will raise my score, otherwise I will retain a **reject**.

## Minor comments
* Some notes on Section 3.1. Why $\epsilon_\theta(x_t, t)$ and not $\mathbb [x_1 | x_t]$ to be more in line with the seemingly preferred notation?
* Likewise I would say that the equation in line 143 is more accurately described as an optimal-transport formulation of flow matching with Gaussian source distribution.
* Moreover, in line 50, why follow $s < t$ for Gaussian flow matching? In flow matching literature we commonly set the source distribution at time 0 and the target at time 1. Some clarity on this would be helpful.
* Especially in light of section 4.4 I would just adopt the flow matching conventions for time (which in the reviewer's opinion are far better and less ambiguous)
* For the work on Section 4.4 the authors should mention some other works which look at gradient guidance and show that the impact is greater at earlier times. While not 100% addressing the same topic these works are closely related and I recommend the authors review them, in particular [1 Section 4] and [2, Proposition 5.2]. These are probably the most relevant theoretical results all there are other heuristic observations from people working on general gradient guidance for flow/diffusion models.

### References
[1] Ben-Hamu, Heli, et al. "D-Flow: Differentiating through Flows for Controlled Generation." International Conference on Machine Learning. PMLR, 2024. https://arxiv.org/pdf/2402.14017

[2] Blasingame et al., "Greed is Good: A Unifying Perspective on Guided Generation", NeurIPS 2025, https://openreview.net/pdf?id=s14pdQgoLb

[3] MOUFAD, Badr, et al. "Variational Diffusion Posterior Sampling with Midpoint Guidance." The Thirteenth International Conference on Learning Representations. https://proceedings.iclr.cc/paper_files/paper/2025/file/ed524bb14de1b52c8522b977ded241d3-Paper-Conference.pdf

[4] He, Yutong, et al. "Manifold Preserving Guided Diffusion." The Twelfth International Conference on Learning Representations.

[5] Pan, Jiachun, et al. "AdjointDPM: Adjoint Sensitivity Method for Gradient Backpropagation of Diffusion Probabilistic Models." The Twelfth International Conference on Learning Representations.

[6] Yu, Jiwen, et al. "Freedom: Training-free energy-guided conditional diffusion model." Proceedings of the IEEE/CVF International Conference on Computer Vision. 2023.

[7] Chung, Hyungjin, et al. "Diffusion Posterior Sampling for General Noisy Inverse Problems." The Eleventh International Conference on Learning Representations.

**Questions:**

1. How is Shielded Diffusion described in equation (3) applied to the ODE solver?
2. What does the *Vendi* score measure?
3. What is a control-barrier function, it's not defined well in the paper. Does it have special mathematical properties over some other map $\mathbb R^d \to \mathbb R$
4. In line 313 what is $L$? Is it a map $\mathbb R \to \mathbb R_{\geq 0}$?
5. In equation (5) why does MMD have the hat symbol over it? Is it is because it is empirically estimated?

---

> ### Author Response · Authors · 2025-11-21
> **Official Comment by Authors**
>
> We thank the reviewer for their thoughtful and detailed comments. We have carefully considered all comments and would like to address the concerns raised.
>
> > ### **Q1 — “How do you pick the time window $S_c$? Isn’t this as heuristic as Safe Denoiser’s interval?”**
>
> We show that negative guidance should begin at the very first sampling step. The termination time $s_c$ is selected through empirical validation across tasks, and the resulting windows consistently lie near the start of denoising, for example [1.0, 0.8] or [1.0, 0.6]. These choices reduce the attack success rate and memorization without degrading fidelity, and the window is fixed per setting rather than tuned per prompt. Please refer to as Figure 4 and Tables 1, 2, and 3.
>
> This empirical pattern is supported by our theory. Subsection 4.4 provides a forward time control barrier certificate, Theorem 2, showing that, for a fixed guidance budget, moving guidance earlier strictly strengthens the safety margin as stated in Equation (10). The same analysis implies that guidance can be turned off after a deadline $s_c$ under a mild forward invariance condition, which specifies when guidance is needed and when it should vanish. We acknowledge that our current analysis is based on the non-explictly known function $h$ and the quantities $\mu$ and $L$. Depending on the applications, however, where one can define $h$, $\mu$ and $L$ explicitly, $s_c$ could be identified by numerically optimizing eq.(10) for $s_c$. While we do not compute $s_c$, the existence of $s_c$ from our theorem motivates the early start used in all experiments and explains why early windows work reliably.
>
> > ### **Q2 — “What is the compute cost of MMD and its gradient? Does it scale poorly with the unsafe reference set?”**
>
> | Models                          | Time (s/img) |
> |---------------------------------|--------------|
> | SD-v1.4                         | 3.18         |
> | + SafeDenoiser (N = 515)        | 3.20         |
> | + Ours (N = 515)                | 3.22         |
> | SAFREE                          | 4.22         |
> | + SafeDenoiser (N = 515)        | 4.24         |
> | + Ours (N = 515)                | 4.32         |
> | + SafeDenoiser (N = 3,200)      | 4.29         |
> | + Ours (N = 3,200)              | 4.70         |
>
> On SD v1.4, SGF with N=515 increases time from 3.18 to 3.22 seconds per image, while Safe Denoiser with N=515 increases time to 3.20. On SAFREE, SGF adds 0.10 seconds for N=515 and 0.48 seconds for $N=3,200$, whereas Safe Denoiser adds 0.02 and 0.07 seconds respectively. We find that the absolute overhead remains as the 1.04 second increase raised by from SD v1.4 to SAFREE. The detailed explanation is updated in the revised manuscript.
>
> This is because our method computes the MMD guidance using a closed form expression rather than automatic differentiation through the sampler. In particular, we use the analytic gradient of the squared kernel MMD in Equation (7). This gradient can be written as a weighted sum of radial terms, which makes it straightforward to evaluate in practice.
>
> In terms of complexity, each negative guidance call has cost that is linear in the size of the unsafe reference set $|D^{-}|$. The computation is implemented as parallel matrix operations on the GPU. Apparently, in our PyTorch implementation, the dominant cost remains the UNet or flow forward passes, and the additional MMD calculations introduce only a minimum overhead. The overall runtime is almost identical to both Safe Denoiser and Shielded Diffusion.
>
> Regarding the size of $D^{-}$, in our experiments a few hundred reference images per domain are sufficient. For the nudity benchmark we use 515 I2P images that exceed the NudeNet threshold, for the diversity experiments we use 50 ImageNet training images per class, and for the memorization experiments we use the ImageNette training split as the negative set. A single unsafe reference image would indeed be brittle, because it reduces the field to a single radial repulsion. In contrast, the moderate coverage described above allows the kernel estimator to produce a stable repulsive signal in the regions of the space that the sampler actually visits.

---

> ### Author Response · Authors · 2025-11-21
> **Official Comment by Authors (2)**
>
> > ### **Q3 — “If Section 4.4 already defines a control‑barrier function $h$, why bother with MMD? Why not standard gradient guidance using $h$?”**
>
> In section 4.4, $h$ is purely theoretical. In some application domains such as robotics, $h$ can be written explicitly from known obstacle geometry and safety margins. However, in image generation, the unsafe set is semantic, and there is ***no simple closed function $h$***. Hence, our method needs a ***data-driven potential***, and our choice is the MMD energy $E(x)$, whose gradient aligns with the intended outward direction by repelling from a finite unsafe reference set. (We will explain why MMD is a good potential to use in next answer.)
>
> When one tries to instantiate a barrier in image generation, the closest construction is Shielded Diffusion (SPELL). Setting $h_i(x)=\lVert x-y_i\rVert-r$ produces a radial, thresholded force that behaves like a ReLU on $r-\lVert x-y_i\rVert$, which is the SPELL rule in Equation (3). This is inherently local because only the nearest point within radius $r$ contributes while distant negatives have little effect. In contrast, our MMD field instead produces a weighted sum of Gaussian radial terms whose magnitude decays exponentially with $\lVert x-y_i\rVert^{2}$, as shown in Equation (7). This concentrates repulsion near genuinely unsafe points and suppresses long range interference, which reduces distortion of benign content. This difference explains the stronger quality and diversity we observe compared with SPELL in Table 2.
>
> > ### **Q4 — “Comparision with previous papers [1–7]**
>
> ***In short, while previous papers mainly focus on attractive guidance to pull trajectories toward desired references or target points by minimizing a distance, our method is explicitly repulsive, using an MMD potential $E(x)$ and a term $+\lambda(t)\nabla_x E(x)$ to push away from the unsafe distribution $D^{-}$.***
>
> The key difference in objective compared with the cited works is that most of previous papers are designed to pull the trajectory toward desired reference points or target distributions, that is, they reduce a certain *point-wise* distance through attractive guidance. In contrast, our method is explicitly repulsive. We define an MMD potential $E(x)$ and add a term of the form $+\lambda(t)\nabla_x E(x)$ to push the sampling trajectory away from the unsafe distribution $D^{-}$. Since the weights $w_i(x)$ are proportional to $k_\sigma(x, y_i)$, points that are close to unsafe exemplars receive stronger repulsive force while far away points have exponentially smaller influence, as can be seen in Equation (7). Thus ***SGF shares the same gradient based mechanism*** as the prior work, but it is applied in ascent form to increase the distance from unsafe data rather than in descent form to move toward a target.
>
> We emphasize that our method utilizes a probability distance potential, directly derived from data, eliminating the need for an additional classifier represented for $h$. This function $E$, being a summation of Gaussians, exhibits *dataset-wise* metrics and demonstrates better stability to sampling from datasets compared to a relatively local approach like SPELL, which can be unstable due to the locality introduced by explicit thresholding. In addition, as we show through Propositions 1 and 2, the same MMD based potential recovers the effective guidance rules of Safe Denoiser and Shielded Diffusion, so SGF provides a single probabilistic view that unifies these earlier negative guidance methods.

---

> ### Author Response · Authors · 2025-11-21
> **Official Comment by Author (3)**
>
> > ### **Q5 — Wouldn’t standard gradient guidance (any of [1–7]) with the barrier function work just as well?”**
>
> ***Shielded Diffusion (SPELL) can be considered as the line of works the reviewer mentioned, but its performance falls short of ours, as evident from Table 2.***
>
> As aforementioned, if we define $h_i(x) = \lVert x - y_i \rVert - r$, this produces a radial, thresholded force that behaves like a ReLU on $r - \lVert x - y_i \rVert$, which corresponds to the SPELL construction in Equation (3). Probably the most serious drawback here is that one has to estimate $h$ from samples, while the above procedure seems to be unstable to sampling. Such a method may work well when the total unsafe data is locally small, but not effective for a largely distributed unsafe data.
>
> In contrast, the MMD potential $E$ used in SGF employs the negative samples directly and induces a field that is a weighted sum of Gaussian radial terms, whose magnitude decays exponentially with $\lVert x - y_i \rVert^{2}$, as shown in Equation (7). As a simple sum of Gaussians, $E$ is stable to the change of the samples, providing a robust method. It also concentrates the repulsive force near genuinely unsafe points and strongly suppresses the influence of distant negatives, which reduces unnecessary distortion of benign content. This difference in how the guidance field is shaped explains the improved quality and diversity trade off that we observe compared with SPELL in Table 2. In some sense, our method takes advantage of the collective effect of the whole negative samples. To reflect this aspect, our method works poorly with small sample size but seems to work well with large samples. Analyzing such a benefit of collective effect in the safety guidance seems to be a promissing future research direction.
>
> > ### **Q6 — Time convention—why set $t=1\to0$ for Gaussian flow matching.**
>
> As the reviewer points out, diffusion models and flow matching typically use opposite time conventions. Our paper covers both diffusion models (SD v1 and v2) and a flow matching model (SD v3), so we had to choose a single convention that can describe all of them in a unified way. For the sampling dynamics, including SD v3, we adopt the diffusion style convention where the source noise distribution is placed at $t = 1$ and the data distribution is placed at $t = 0$. This matches the probability flow ordinary differential equation view of diffusion models and reflects how SD v3 is actually implemented in the official code. From a practical point of view, using a common index for all samplers helps readers who look at the implementation avoid confusion about whether time is increasing or decreasing.
>
> For the safety analysis in Section 4.4, we introduce a separate, purely analytic time variable that runs forward from 0 to 1. In that section we set the initial time to 0 and the final time to 1 in order to match the usual convention in control barrier function theory, where the dynamics evolve forward in time and the barrier condition is written as a forward invariance constraint. This analytic time is only used for the derivation of Theorem 2 and for expressing the margin and inequality in Equation (10) in a clean way with all terms appearing with positive sign. It does not affect the actual sampling algorithm or the pseudocode, which continue to use the implementation time index described in the previous paragraph.
>
> We update this explicit by adding a short note on time conventions at the subsection 3.1 in the revised manuscript. We state clearly that the implementation uses a unified diffusion style index with source at t = 1 and target at t = 0 for both diffusion and flow matching samplers, while the analytic control barrier argument uses a separate forward time variable in [0, 1] only for theoretical clarity.

---

> ### Author Response · Authors · 2025-11-21
> **Official Comment by Authors (4)**
>
> ## **Minor Comments**
>
> > **Please relate your Section 4.4 analysis to prior theory on gradient guidance that shows guidance has stronger effects at earlier times, especially D-Flow [2, Sec. 4] and Greed-is-Good [3, Prop. 5.2].**
>
> Both [2] and [3] directly tackle the question of how best to approximate the true end-to-end guidance gradient. Greed-is-Good analyzes greedy and posterior updates as low-order approximations to this gradient, while D-Flow studies differentiating through the flow as a principled but computationally costly approximation scheme.
>
> On the other hand, our paper proposes a closed-form, training-free potential (MMD) and studies safety scheduling and policy. In this paper, we answer when a repulsive guidance field should be active. Our theorem on the “critical early window” is orthogonal to the approximation issues in [2,3].
>
> However, our theorem supposes that the proposed MMD gradient guidance aligns with the ideal control barrier field near the boundary. The revised manuscript explicitly states in the new limitation paragraph at Section 6 that future analysis of approximate safety guidance is promising and we cite these two papers in that context.
>
> > **How is Shielded Diffusion in Eq. (3) applied inside the ODE solver?**
>
> In our implementation, SPELL acts on the model’s predicted clean sample $z_t=\mathbb{E}[x_0\mid x_t]$. At each step we compute the radial, thresholded force in Eq. (3) for all references and form the corrected clean target $z_t^{\text{SPELL}} = z_t + \sum_j F_{\text{rad}}(z_t; y_j)$. We then update the state using the standard Euler ODE step while replacing the data‑prediction term $\mathbb{E}[x_0\mid x_t]$ in Eq. (2) by $z_t^{\text{SPELL}}$. Appendix C makes this explicit by applying the force in clean space and steering the sampler through the corrected target.
>
> > **What does the Vendi score measure?**
>
> Vendi is a diversity metric that quantifies the effective number of distinct samples in an embedding space through the entropy of a similarity kernel’s spectrum [1]. Higher Vendi indicates a set with greater variety, while a value near one indicates low diversity.
>
> > **Q: What is a control‑barrier function and does it have special mathematical properties?**
>
> In our analysis a control‑barrier function is a continuously differentiable map $h:\mathbb{R}^d\to\mathbb{R}$ whose nonnegative superlevel set $S=\{h\ge 0\}$ encodes safety. Section 4.4 states conditions under which the guidance $\beta(s)\nabla E$ aligns with $\nabla h$ and the base drift $\tilde f$ is bounded near the boundary. Under these conditions Theorem 2 shows that a finite amount of early guidance produces a positive margin by a deadline $s_c$. This is a forward‑invariance type property tailored to our setting and is what we use to formalize the early critical window.
>
> > **Q: In line 313, what is the symbol $L$ and is it a map ?**
>
> At line 313 we use the same $L$ defined in Assumption 1(b). A measurable function $L:[0,1]\to\mathbb{R}_{\ge 0}$ that bounds the barrier‑normal component of the base drift in the boundary layer via $|\nabla h(x)\cdot \tilde f(s,x)|\le L(s)\,|h(x)|$. This $L$ induces the decreasing weight $\bar w_L(u)=\exp(\int_u^{s_c}L(\tau)\,d\tau)$ and the weighted guidance mass $\bar I_L(s_c)=\int_0^{s_c}\bar w_L(u)\beta(u)\,du$ in Eq. (9), which are then used in Theorem 2.
>
> > **Q: In Eq. (5), why is there a hat on MMD?**
>
> The hat indicates the empirical squared MMD computed from a finite unsafe reference set. As written in the main text, we use the biased estimator due to the different number of indices in summation, and spell out its finite‑sum form, which is what we differentiate to obtain Eq. (7).
>
> > **Minor notation in Section 3.1.**
>
> We already use $z_t\equiv \mathbb{E}[x_0\mid x_t]$ and will state this equivalence once and then consistently use $\hat x_{0|t}$ in Section 3.1 while keeping $z_t$ in algorithms for brevity.
>
> **Reference**
>
> [1] Friedman, Dan, and Adji Bousso Dieng. "The vendi score: A diversity evaluation metric for machine learning." TMLR2023
>
> [2] Ben-Hamu, Heli, et al. "D-flow: Differentiating through flows for controlled generation." ICML2024
>
> [3] Blasingame, Zander W., and Chen Liu. "Greed is Good: A Unifying Perspective on Guided Generation." NeurIPS2025

---

> > ### Comment · Reviewer_UsAs · 2025-11-22
> >
> > The authors have made great efforts to improve the clarity of their paper. The changes made in their pdf answer most of my questions.
> >
> > I only have two remaining questions/suggestions.
> >
> > For **Q1** I feel the authors could expand more about how they derive a theoretical justification for selecting $S_c$ and in particular a practitioners guide on how to find it. When I reread through the paper it seemed to be missing, if I just missed just please comment the section where it's located.
> >
> > For **Q3** it looks like $h$ is a scoring function for how "safe/unsafe" an image is. Could you not just train a neural network and use this scoring function explicitly? What are the advantages of using a dataset?

---

> ### Author Response · Authors · 2025-11-22
> **Response to Reviewer UsAs's comment**
>
> We appreciate the reviewer’s prompt feedback. We have carefully examined the remaining concerns and provide our responses below.
>
> > ### **Q1. Updating the manuscript for the limitation on $s_c$.**
>
> We update a limitation regarding the selection of $s_c$ in Section 6. We explicitly state as follows. *"Although our theory certifies the existence of an early critical window, the representative windows used in our experiments, such as [1.0, 0.8]  and [1.0, 0.6], are selected through empirical validation across tasks rather than computed from the theoretical analysis. Developing a principled, data driven estimator for $s_c$ is an important future direction."*
>
> > ### **Q2. Advantages of data-driven negative guidance over a trained classifier.**
>
> ***One could train a neural network and use its score surrogated for $h$, but a data-driven MMD potential guidance is preferred because it requires no training, works well with few examples, is transparent and easy to update, and generalizes concepts and base-models with minimal efforts.***
>
> A pre-trained classifier can theoretically play the role of the function surrogated for $h$, and our framework acknowledges this choice. If a reliable, differentiable safety model already exists for a specific category, it can be utilized for guidance [1].
>
> However, we focus on the data-driven MMD potential guidance for image generation because it is training-free, computationally light, model agnostic, and adaptable. Not every safety concept has a robust classifier. Many high level or sensitive categories, such as artist style or narrow intellectual property cases, lack a trustworthy pretrained classifier. Training a new network for such a concept often requires more than a handful of data points. With very small sets, a classifier tends to overfit, resulting in unstable gradients that can misguide the generation trajectories. In contrast, MMD immediately converts any available negative data points into a repulsive field as shown in Appendix F.2. It works well with modest reference sets and can be updated by simply adding or removing examples without training additional classifiers.
>
> From a deployment perspective, this approach reduces efforts for maintenance. Service providers no longer need to train and monitor separate classifiers for every policy change. Instead, they can curate or accept curated negative datapoints from domain experts and take this policy immediately through MMD guidance. Therefore, the data-driven negative guidance remains a simple and straightforward option that adapts across models, concepts with minimal engineering effort.
>
> **Reference**
>
> [1] Na, Byeonghu, et al. "Training-Free Safe Text Embedding Guidance for Text-to-Image Diffusion Models." NeurIPS2025.

---

> > ### Comment · Reviewer_UsAs · 2025-11-23
> >
> > I have no further questions and will update my score accordingly.

---

> ### Author Response · Authors · 2025-11-23
> **Response to Official Comment by Reviewer UsAs**
>
> We sincerely appreciate the reviewer for their constructive and fruitful discussion. We ensure that all comments enhance the clarity of our paper.

---

### Official Review · Reviewer_TunD · 2025-11-01

**Soundness:** 3
**Presentation:** 3
**Contribution:** 3
**Rating:** 6
**Confidence:** 3

**Summary:**

The paper presents a unified MMD-based formulation of negative guidance for safety-aware generation. The authors show that this new MMD-based framework generalizes previous arts, Shielded Diffusion and Safe Denoiser. Moreover, they provide a control barrier function analysis, finding that the guidance strength must decay so as to remove undesirable nonlocal guidance impact on already safe regions. The proposed method, SGF, is evaluated on several carefully designed experiments, showing that it can effectivively prevent offensive content generation without hurting too much on diversity. Additionally, SGF is also shown to be effective at mitigating the memorization issue.

**Strengths:**

The theoretical insights are novel. The authors provide a more principled objective for safety-aware negative guidance. Unlike previous methods' formulation based on binary/proximity classification, the proposed SGF views the problem as maximizing a proper divergence metric (MMD) between the undesirable distribution and the generated distribution. The critical time window theory also explains why early stopping is effective.

**Weaknesses:**

While the theoretical insights are novel, the pragmatic novelty is limited. The paper is mainly focused on 'why it works.' For instance, the MMD-based formulation is sound and novel, but the resulting parametric form of the guidance model itself is effectively identical to SafeDenoiser. The critical time window theory provides why certain stopping parameter is better, but this can be empirically chosen without theory.

**Questions:**

None.

**Details Of Ethics Concerns:**

The paper contains censored version of nudity images with explicit prompts. The authors appropriately warn this in the beginning of the paper.

---

> ### Author Response · Authors · 2025-11-21
>
> We appreciate the invaluable comments. We have carefully considered all comments and would like to address the concerns raised.
>
> > ### **Novelty and safety guarantees**
>
> As reviewer h63Q already emphasized, our paper provides the first formal safety guarantees for generative flows through a control barrier certificate and by unifying previously empirical methods within one MMD potential framework. This is more than a conceptual consolidation. It yields concrete test time policies that measurably improve the safety and quality trade off across samplers and backbones.
>
>
> > ### **Principled MMD energy and unified view**
>
> First, SGF is not a rebranding of Safe Denoiser. We prove that the practical repellency used in Safe Denoiser is the gradient of an explicit MMD energy evaluated at the predicted clean sample $z_t$, up to a scale that depends on the bandwidth and on the local kernel normalizer $Z(z_t)$. This turns an implementation detail into a principled potential, specifies which normalization matters, and clarifies how guidance strength should adapt to the local kernel mass. The equivalence is stated in Proposition 1 and Appendix B.1, with the closed form gradient in Equation 7 and the identity $g_{\text{SD}}(t)=\frac{\sigma^2}{2Z(z_t)}\nabla_{z_t}E(z_t)$. This probabilistic view was not available to Safe Denoiser. It also lets us relate Shielded Diffusion(SPELL) to the same potential through a radius to bandwidth matching argument, so both become special cases of one energy based negative guidance.
>
> > ### **Critical window certificate: when to apply guidance and when to stop**
>
> Second, the critical window certificate answers the practical question that prior work left empirical, namely when to apply guidance and when to turn it off. Theorem 2 shows that under the standard image and video schedules where the base drift contracts near the end, the sufficient bound places more weight on earlier guidance through a decreasing exponential factor and yields a constructive rule to stop guidance after a deadline to preserve fidelity. This directly predicts that earlier is better and that guidance should stop once the sample is safely separated. Our ablation isolates timing from strength. Figure 4(c) holds amplitude and window length fixed and only shifts the window. It produces the ranking predicted by the theorem, with the earliest windows [1.0,0.8] and [0.8,0.6] attaining the lowest ASR across all three red teaming sets and performance degrading as the same width window is moved later. Panels 4(a) and 4(b) reproduce the same conclusion under the more confounded settings of varying length and equal budget. This is a practical policy with a formal certificate, not only an empirical rule.
>
> > ### **Summary**
>
> In short, SGF goes beyond explaining an existing heuristic. It formalizes negative guidance as energy minimization, proves equivalences that unify two previously separate lines of work, derives a certified early stop policy that changes how guidance is scheduled, and demonstrates measurable gains across adversarial nudity, memorization, and a flow matching backbone. These points address the concern about limited pragmatic novelty while aligning with reviewer h63Q’s assessment that our cross disciplinary analysis provides the first formal safety guarantees for generative flows and that the experiments validate the theory.
>
> > ### **Ethical considerations**
>
> In the initial manuscript, we included a red-highlighted warning immediately after the abstract on the first page: “Warning: This paper contains disturbing content such as sexually explicit images and words.”
>
> To draw more attention to this warning, we have revised it in the updated manuscript to “Warning: This paper contains disturbing content, including censored images of nudity and sexually explicit text prompts, presented for research purposes only.”

---

### Official Review · Reviewer_ATH6 · 2025-11-01

**Soundness:** 3
**Presentation:** 3
**Contribution:** 2
**Rating:** 6
**Confidence:** 3

**Summary:**

This work proposes a unified framework for diffusion models with negative guidance for safe generation. The authors apply Maximum Mean Discrepancy (MMD) as a potential function to measure the distance between the current generation and a negative sample dataset, then use its gradient as guidance for the diffusion generation process. By choosing different kernel functions, the framework recovers existing works such as Shielded Diffusion and Safe Latent Diffusion. The paper further investigates the effectiveness of applying guidance at the early stages of the generation process and demonstrates this through empirical studies.

**Strengths:**

1. Safe generation is critical for real-world applications, making this an important research direction.
2. The proposed unified framework based on MMD guidance effectively covers and connects recent works in the field.
3. The early-stage guidance analysis is insightful and validated through empirical studies.

**Weaknesses:**

1. Assumption 1(b) needs more justification. While the authors provide an intuitive understanding for the assumption at the final time step, it is unclear how this assumption holds at other time steps and how it should be interpreted more generally. If this is a standard choice in the control barrier function literature, please provide a detailed discussion and relevant citations.
2. Theorem 2 and the ablation study need better alignment and justification. Why does the ASR decrease and then increase across the three time window settings? How does this non-monotonic behavior align with the analysis provided in Theorem 2? In practice, what is the best approach to set the guidance window and budget based on the analysis of Theorem 2?

**Questions:**

1. How do the negative samples in the potential function calculation impact generation? Specifically, what are the effects of the number and diversity of negative samples on the final output quality and safety?
2. Theoretically, how can we understand the tradeoff between generation quality (distribution approximation error) and safety (constraint satisfaction) under the negative guidance framework?
3. Despite the computational efficiency of the MMD potential function and its ability to cover two recent works, what are the potential drawbacks compared to using other probability measures as the potential function?


I will adjust my score upon clarification of these points.

**Details Of Ethics Concerns:**

This paper contains disturbing content, such as sexually explicit images and words, as mentioned in the abstract.

---

> ### Author Response · Authors · 2025-11-21
> **Official Comment by Authors**
>
> We thank the reviewer for their thoughtful and detailed comments.
>
> > ### **Q1 — Justification of assumption 1(b) and literature survey of control-barrier function**
>
> As reviewer h63Q noted, assumption 1(b) concerns the underlying generative dynamics driven by the drift $\tilde f$. As we noted in the paper, we expect that proximity to the unsafe region primarily near the final reverse time of the diffusion or flow matching process, where the drift $\tilde f$ is expected to be small. For instance, in diffusion models, denoising is still in progress before the final time, so at intermediate times the state remains noisy and typically maintains a certain distance from harmful content.
>
> Furthermore, this assumption is standard in the control barrier function (CBF) literature. In the manuscript, we have already cited the relevant sources [1-3] at the beginning of Subsection 4.4. Here, we summarize the key statements for clarity. Nguyen and Koushil [1] consider a CBF setting as a one dimensional safe case of the form $\dot B \ge - k_b B$, which leads directly to the exponential bound $B(t) \ge B(0)\mathrm e^{-k_b t}$. This appears in Definition 2
> together with the surrounding discussion in [1]. Our work follows this setting.
> Glotfelter et al. introduce the nonsmooth CBF setting that  is a continuous and strictly increasing function, and they propose a generalized one dimensional safe differential inequality [2].
>
> SafeDiffuser[3] targets robotic planning with a diffusion planner. While this work incorporated CBF based guidance into diffusion or flow sampling, it applies this guidance at every denoising step and does not analyze when it is actually necessary.
>
> In conclusion, as reviewer h63Q noted, our contribution is to provide an analytic justification of an early stop strategy for negative guidance within these classical CBF theory and to show that in the SGF setting these constraints yield a closed form critical window certificate. This demonstrates that the theoretical analysis and the proposed guidance schedule are tightly aligned. Regarding relaxation of Assumption 1(b), a more rigorous analysis of this behavior would be a good research direction.
>
> > ### **Q2 — Alignment of Theorem 2 and ablation study**
>
> Figure 4.c is the clearest one to one test of Theorem 2. Panel c fixes the per step guidance strength within the window and keeps the window length constant, and only shifts this same window earlier or later along the trajectory. Under this design, the integrated guidance budget is identical across all settings, so the only factor that changes is the timing of when that budget is used. Theorem 2 gives a sufficient certificate in which earlier guidance is weighted more heavily through the exponentially decaying factor $\exp \big(\int_{u}^{s_c} L \big)$, so concentrating the same guidance mass earlier is predicted to yield the strongest safety effect. This is exactly what we observe in Figure 4.c. The earliest windows [1.0, 0.8] and [0.8, 0.6] achieve the lowest ASR across all three red teaming benchmarks, and the ASR worsens monotonically as the fixed width window is shifted later. Because amplitude and length are held fixed and only timing is varied, this panel provides a direct, controlled confirmation of the prediction of Theorem 2.
>
> Fairness checks with the other two ablations. Panels a and b explore more entangled regimes, where we vary the window length at fixed amplitude and use equal budget schedules that scale amplitude with length. Even in these settings, the best ASR is obtained when guidance is placed early, with [1.0, 0.8] and [1.0, 0.6] again outperforming later windows across datasets. The same pattern appears for Safe Denoiser in Appendix Figure E.2, which repeats all three ablations and reproduces the “early best, late worse” trend. These two panels therefore act as consistency checks that support the primary timing only test in panel c.
>
> Why the ASR curve can worsen when the window intrudes into late steps: The non monotonic behavior noted by the reviewer, where ASR first decreases and then increases as the window extends into late refinement steps, arises when an initially early window is lengthened into the final part of denoising. Early guidance mass helps the trajectory leave unsafe regions, but additional late guidance can distort the fine grained refinement phase and push samples toward regions where the model is less well trained. Our two dimensional toy study illustrates this effect visually. When negative guidance is applied throughout, a noticeable population remains near the unsafe set and neighboring modes are distorted, whereas early stop guidance both reduces mass near the unsafe region and better matches the target distribution, with Wasserstein distance improving from $W^2 = 1.009$ to $W^2 = 0.937$. This explains why extending guidance into the tail end of the trajectory can eventually worsen ASR after an initial gain. The side by side outcomes are shown in Figure 2(c–d)

---

> ### Author Response · Authors · 2025-11-21
> **Official Comment by Authors (2)**
>
> > ### **Q3 — How do the negative samples in the potential function calculation impact generation? Specifically, what are the effects of the number and diversity of negative samples on the final output quality and safety?**
>
> Prior work, Safe Denoiser [4], has already analyzed this question in detail. In the Safe Denoiser paper, the set size ablation in Figure 4(a) reduces the negative pool by subsampling and shows that ASR increases as the set becomes smaller. When the set is very small such as around 100 negatives, the ASR curve approaches that of a text pre-filter approach such as SAFREE. This ablation implicitly reflects sensitivity to both the number of negatives and their effective coverage and diversity, because the size is changed by subsampling from a common pool. Table 1 in that paper provides the full set reference baseline for comparison.
>
> We therefore expect SGF to show the same qualitative behavior. Since this perspective has already been established in Safe Denoiser [4], and in order to keep the core message of our paper focused, we chose to cite those results rather than repeat the same ablation in the main text. Furthermore, we emphasize that the need for negative examples is not a special drawback of our approach. Pre-filter and post-filter methods also rely on curated data, while Safe Denoiser uses negatives at inference time without retraining and can enhance safety when combined with text guards.
>
> At the method level, the method closest to using a control barrier function in robotics is Shielded Diffusion (SPELL). It checks the nearest negative points, applies a radial threshold and mirrors a CBF inside gradient guidance. If we define per‑exemplar barriers $h_i(x)=( \lVert x-y_i\rVert - r)^2$, the standard gradient rule yields a radial, thresholded force that behaves like a ReLU on $r-\lVert x-y_i\rVert$ and matches the SPELL construction in Equation (3). This is inherently local because only the nearest point within radius $r$ contributes while distant negatives have little effect. Since Assumption 1(a) concerns dynamics near the boundary, its locality can bias the estimate of $h$, which weakens the certificate.
>
> In contrast, SGF builds a global potential from all negatives and steers sampling by descending that energy. We use the MMD potential $\Phi_{\text{MMD}}(x)=\tfrac{1}{n}\sum_j k_\sigma(x,y_j)$ with the RBF kernel $k_\sigma(x,y)=\exp \big(-\lVert x-y\rVert^2/(2\sigma^2)\big)$ and apply guidance through $-\nabla_x\Phi_{\text{MMD}}(x)$, where $\nabla_x k_\sigma(x,y)=-(x-y)k_\sigma(x,y)/\sigma^2$. The resulting field aggregates distance‑aware contributions from both near and far points, strengthens near the boundary, and decays smoothly at long range, so it captures global unsafe density and boundary geometry without fitting $h$ explicitly. This reduces sensitivity to any single neighbor and aligns with Assumption 1(a) when boundary coverage is modest. Consistent with this, Table 2 shows that SGF achieves smaller FID loss while achieving stronger diversity than SPELL, indicating that SGF removes unsafe mass while describing the target distribution with less distortion.

---

> ### Author Response · Authors · 2025-11-21
> **Official Comment by Authors (3)**
>
> > ### **Q4 — Theoretically, how can we understand the tradeoff between generation quality (distribution approximation error) and safety (constraint satisfaction) under the negative guidance framework?**
>
> Safety constraints partition the data distribution $P_{\text{data}}$ into safe and unsafe regions. Enforcing the constraint means removing or relocating the unsafe mass $\pi$. A natural way to formalize this is through unbalanced optimal transport, where transport and mass creation/annihilation are allowed. In the Wasserstein–Fisher–Rao geometry, the loss in quality corresponds to the geodesic cost betweon the orignal distribution and its safe modification; the cost simultaneously accounts for two effects, namely, displacement, captured by the Wasserstein part, and reaction, captured by the Fisher–Rao part.
>
> Within this framework, the role of negative guidance strength becomes clear. When we apply negative guidance, the parameter $\lambda$ increases the penalty associated with leaving probability mass in unsafe regions. Conceptually this plays a role similar to increasing the weight on mass change in unbalanced optimal transport. As $\lambda$ becomes larger, the amount of unsafe mass decreases, but at the same time the overall distribution moves farther away from $P_{\text{data}}$. In other words, varying $\lambda$ traces out a Pareto frontier between safety and fidelity. Static formulations of unbalanced transport make this explicit by placing a tunable weight on divergence terms that control how much mass can change.
>
> If the unsafe proportion is $\pi$, then any fully safe target distribution must at least relocate that entire amount $\pi$, either through transport or through reaction. Except in degenerate geometries where transport is effectively free, unbalanced transport therefore predicts a nonzero lower bound on quality loss that grows with $\pi$ and with the cost of transport and reaction. Pushing $\lambda \to \infty$ forces safety as much as possible, but in the limit it becomes maximal such a lower bound of the cost of fidelity.
>
> > ### **Q5 — Despite the computational efficiency of the MMD potential function and its ability to cover two recent works, what are the potential drawbacks compared to using other probability measures as the potential function?**
>
> An MMD potential provides a computationally light, repulsive guidance field that is aligned with the behavior targeted by prior safety methods such as Shielded Diffusion and Safe Denoiser.
>
> Using $W_2^2$ directly as the potential would amount to maximizing $W_2^2(\mu_{\text{curr}},\mu_{\text{unsafe}})$. However, $W_2^2$ is a distribution level objective whose gradient propagates through the optimal coupling or Kantorovich potential rather than through simple pairwise $\|x - y\|^2$. In practice, this requires re-training for flow matching for unsafe distrubition or training at each step and therefore produces a global signal that is not well matched to our goal of training free, test time scaling of safety guidance.
>
> For this reason, we use an empirical RKHS potential $\Phi_{\text{MMD}}(x) = \tfrac{1}{n}\sum_j k_\sigma(x, y_j)$ built from negative datapoints $y_j$, and we apply negative guidance through the repulsive field, $\nabla_x \Phi_{\text{MMD}}(x)$. For an RBF kernel $k_\sigma(x,y) = \exp(-\|x-y\|^2/(2\sigma^2))$, the gradient is $\nabla_x k_\sigma(x,y) = -(x-y) k_\sigma(x,y)/\sigma^2$ and its magnitude is $\|\nabla_x k_\sigma\| = \|x-y\|\,\exp(-\|x-y\|^2/(2\sigma^2))/\sigma^2$. The guidance is zero at coincidence, reaches its maximum near $\|x-y\| \approx \sigma$, which defines a finite safety radius, and decays exponentially at larger distances. This yields strong local repulsion from unsafe exemplars with negligible influence once samples are already far away, which mirrors the behavior of prior safety guided samplers without requiring additional training for new optimal transport.
>
> The main limitation of the MMD potential is sensitive to the bandwidth $\sigma$. If the bandwidth is chosen poorly, the barrier can become either overly smooth or too myopic. The problem of choosing a proper bandwidth is simlar to the problem of choosing a proper shield radius in Shielded Diffuions. We empirically mitigate this by adaptive bandwidths as described in Appendix D. Furthermore, since the empirical MMD is based on pairwise sums over negatives and current samples, the resulting gradients can be noisy and is prone to a poor unsafe datapoints sampling. This drawback is also shared by Safe Denoiser and Shielded Diffuions.

---

> ### Author Response · Authors · 2025-11-21
> **Official Comment by Author (4)**
>
> > ### **Ethical considerations**
>
> In the initial manuscript, we included a red-highlighted warning immediately after the abstract on the first page: “Warning: This paper contains disturbing content such as sexually explicit images and words.”
>
> To draw more attention to this warning, we have revised it in the updated manuscript to “Warning: This paper contains disturbing content, including censored images of nudity and sexually explicit text prompts, presented for research purposes only.”
>
> **Reference**
>
> [1] Nguyen, Quan, and Koushil Sreenath. "Exponential control barrier functions for enforcing high relative-degree safety-critical constraints." 2016 American Control Conference (ACC). IEEE, 2016.
>
> [2] Glotfelter, Paul, Jorge Cortés, and Magnus Egerstedt. "Nonsmooth barrier functions with applications to multi-robot systems." IEEE control systems letters 1.2 (2017): 310-315.
>
> [3] Xiao, Wei, et al. "Safediffuser: Safe planning with diffusion probabilistic models." ICLR2025.
>
> [4] Kim, Mingyu, et al. "Training-free safe denoisers for safe use of diffusion models." NeurIPS 2025.

---

### Author Response · Authors · 2025-12-03
**General Response by Authors**

We sincerely thank all the reviewers for their insightful and constructive feedback on our manuscript. Every comment has been invaluable to our work.

---

### **List of major concerns we addressed in the rebuttal**

1. **Theoretical explanation on Assumption 1 & Control-barrier function(CBF)** : We clarified the role of Assumption 1 and Theorem 2 and connected our setting to standard control barrier function theory. → *reviewers ATH6, h63Q*

2. **Critical early time window from both theoretical and practical analysis** : We aligned Theorem 2 with the ablation studies, highlighting that Figure 4(c) holds guidance strength and window length fixed and only shifts the window, thereby directly testing the theorem that earlier guidance is more effective. → *reviewers ATH6, TunD, UsAs, h63Q*

3. **Novelty** : We prove that the practical repulsive field used in `Safe Denoiser`[1] is the gradient of an explicit kernel MMD potential. We also show that `Shielded Diffusion`[2] can be recovered from the same potential via a radius–bandwidth correspondence.  → *reviewers ATH6, TunD, UsAs, h63Q*

4. **Data-driven training-free negative guidance** : We argued that a data-driven MMD potential is training-free, transparent, easy to update, and works well with modest negative sets → *reviewers UsAs, ATH6, h63Q*

5. **Computational overhead and scalability with the unsafe reference set** : We reported detailed wall-clock time showing that SGF introduces only modest overhead compared with the base sampler (SDv1.4) and with SAFREE, even when the negative set size increases (e.g., from $N = 515$ to $N = 3,200$). → *reviewers UsAs, h63Q*

6. **Applicability beyond nudity** : We explained that SGF is compatible with more abstract concepts like copyright and style, and practically connected our method to intellectual property. → *reviewer h63Q*

---

We have thoroughly addressed all issues raised by the reviewers, both through the rebuttal and via revisions to the main paper and supplementary material. In particular, *reviewer UsAs*, who initially leaned toward rejection, stated after reading our responses that all remaining questions had been resolved and explicitly indicated that he raises his score. The most positive *reviewer h63Q* likewise confirmed that our answers fully addressed his concerns and that no issues remain. *Reviewer ATH6* had also noted in the original review that he would adjust his score once his questions were clarified. Although the updated ICLR 2026 policy no longer allows a follow‑up comment and score change, our responses directly resolve the points he raised, and we are confident that Reviewer ATH6's evaluation is aligned with the other reviewers’ updated positions.

Once again, we sincerely thank the reviewers, AC, and SAC for their thoughtful feedback and engagement, which have substantially improved the clarity and scope of our work.

**Reference**

[1] Kim, Mingyu, et al. "Training-free safe denoisers for safe use of diffusion models." NeurIPS2025.
[2] Kirchhof, Michael, et al. "Shielded Diffusion: Generating Novel and Diverse Images using Sparse Repellency.", ICML2025.

---

### Meta-Review · Area_Chair_Z1By · 2026-01-07

**Summary:**

Overall, the reviewers’ scores are **on the higher side** and broadly consistent with acceptance. The paper is viewed as **technically strong**, with clear theoretical contributions (explicit MMD potential, unification of prior methods, and a control-theoretic certificate) and solid empirical validation. The main concerns—window selection, assumptions in the theory, scalability, and novelty—were largely addressed in the rebuttal, leaving only **minor residual limitations** (e.g., empirical choice of the cutoff time and dependence on representative negative data).

My remaining hesitation is not about correctness or quality, but about **audience breadth**: the work is most impactful for researchers at the intersection of diffusion models and safety, rather than the full ICLR community. Balancing this, I am **inclined to recommend acceptance**, and lean toward **oral** in line with the strongest reviewer’s assessment, but my preference between **oral and poster is not strong**. I am comfortable deferring to the **SAC’s final placement decision**.

**Reviewer Concerns:**

**Addressed in rebuttal / revision**

* **Compute cost & scalability**: Provided concrete wall-clock timings and clarified the **closed-form MMD gradient** with **linear scaling in (|D^-|)** and GPU batching; overhead is modest relative to UNet/flow passes.
* **Why MMD vs. “just use a CBF score (h)” / standard gradient guidance**: Clarified that (h) is **theoretical** here (no closed-form semantic safety function for images), and MMD gives a **training-free, data-driven** potential; also contrasted with SPELL’s more **local/thresholded** behavior and showed better quality–safety trade-offs empirically.
* **Assumption 1(b) justification**: Better connected to **standard CBF literature**, and explained how the “small drift near the end” regime matches common image/video schedules.
* **Alignment of Theorem 2 with ablations**: Pointed to the “timing-only” ablation (fixed strength + fixed window length, only shifted) as a direct test; explained the **non-monotonic** patterns when guidance intrudes into late refinement.
* **Edge cases / “can drift overpower guidance?”**: Acknowledged this is the boundary of assumptions; argued why the **kernel field** helps and included stress-test discussion (hard benchmarks still see large relative reductions).
* **Beyond nudity**: Added/clarified evidence and discussion that the framework can handle **more abstract harms** (e.g., copyright/style) via negative exemplars.

**Still outstanding / partially outstanding**

* **Choosing the cutoff (s_c) in a non-heuristic way**: While Theorem 2 motivates “early is better” and provides a certificate-style explanation, **(s_c) is still selected empirically**; there isn’t a practical estimator that reliably computes it from the theory.
* **Sensitivity to negative set quality/diversity**: The paper acknowledges dependence on a representative unsafe set and references prior ablations; but a **dedicated, systematic sensitivity study** (beyond what’s already cited/added) remains limited.
* **Novelty perception**: The rebuttal strengthens the case (explicit MMD potential + unification + certified scheduling), but for at least one reviewer the **pragmatic novelty** concern is only **partially** resolved because the resulting guidance looks similar to Safe Denoiser in the image instantiation.

**Reviewer Scores:**

* **Reviewer h63Q**: **No change (10 → 10)**. All concerns were fully addressed; the reviewer explicitly confirmed satisfaction and already gave the highest score.

* **Reviewer UsAs**: **Increase (4 → ~7)**. Core concerns on window selection, compute cost, and MMD vs. barrier guidance were addressed with theory, concrete timings, and clear justification; the reviewer stated they would raise their score.

* **Reviewer TunD**: **Likely unchanged (6 → 6)**. Theoretical contributions were acknowledged, but the pragmatic novelty concern remains partially; no indication of a score change.

* **Reviewer ATH6**: **Increase (6 → ~7)**. Assumptions, theory–ablation alignment, and practitioner guidance were clarified; some heuristic aspects remain, but overall confidence improved.

---

### Decision · Program_Chairs · 2026-01-26

Accept (Oral)